# Melatonin Exerts Prominent, Differential Epidermal and Dermal Anti-Aging Properties in Aged Human Eyelid Skin Ex Vivo

**DOI:** 10.3390/ijms242115963

**Published:** 2023-11-04

**Authors:** Tara Samra, Tatiana Gomez-Gomez, Kinga Linowiecka, Aysun Akhundlu, Gabriella Lopez de Mendoza, Matthew Gompels, Wendy W. Lee, Jennifer Gherardini, Jérémy Chéret, Ralf Paus

**Affiliations:** 1Dr. Phillip Frost Department of Dermatology & Cutaneous Surgery, University of Miami Miller School of Medicine, Miami, FL 33125, USA; tsamra@med.miami.edu (T.S.); txg548@miami.edu (T.G.-G.); kxl1082@med.miami.edu (K.L.); axa3549@med.miami.edu (A.A.); jxg5462@med.miami.edu (J.G.); jpc219@med.miami.edu (J.C.); 2Department of Human Biology, Faculty of Biological and Veterinary Sciences, Nicolaus Copernicus University, Lwowska 1, 87-100 Torun, Poland; 3Bascom Palmer Eye Institute, Department of Ophthalmology, University of Miami Miller School of Medicine, Miami, FL 33125, USA; 4Monasterium Laboratory, 48149 Muenster, Germany; 5CUTANEON—Skin & Hair Innovations, 22335 Hamburg, Germany

**Keywords:** skin aging, melatonin, p-S6, VEGF-A, TFAM, VDAC/Porin, collagen 17A1, MMP-1, fibrillin-1, SIRT-1, p16INK4, mTORC1

## Abstract

Human skin aging is associated with functional deterioration on multiple levels of physiology, necessitating the development of effective skin senotherapeutics. The well-tolerated neurohormone melatonin unfolds anti-aging properties in vitro and in vivo, but it remains unclear whether these effects translate to aged human skin ex vivo. We tested this in organ-cultured, full-thickness human eyelid skin (5–6 donors; 49–77 years) by adding melatonin to the culture medium, followed by the assessment of core aging biomarkers via quantitative immunohistochemistry. Over 6 days, 200 µM melatonin significantly downregulated the intraepidermal activity of the aging-promoting mTORC1 pathway (as visualized by reduced S6 phosphorylation) and MMP-1 protein expression in the epidermis compared to vehicle-treated control skin. Conversely, the transmembrane collagen 17A1, a key stem cell niche matrix molecule that declines with aging, and mitochondrial markers (e.g., TFAM, MTCO-1, and VDAC/porin) were significantly upregulated. Interestingly, 100 µM melatonin also significantly increased the epidermal expression of VEGF-A protein, which is required and sufficient for inducing human skin rejuvenation. In aged human dermis, melatonin significantly increased fibrillin-1 protein expression and improved fibrillin structural organization, indicating an improved collagen and elastic fiber network. In contrast, other key aging biomarkers (SIRT-1, lamin-B1, p16INK4, collagen I) remained unchanged. This ex vivo study provides proof of principle that melatonin indeed exerts long-suspected but never conclusively demonstrated and surprisingly differential anti-aging effects in aged human epidermis and dermis.

## 1. Introduction

Skin aging results from the combination of many factors and pathways, including oxidative stress, photodamage, mitochondrial dysfunction, DNA damage, telomere shortening, increased mTORC1 and reduced sirtuin-1 activity, extracellular matrix degradation, and a proinflammatory senescence-associated secretory phenotype, to name but a few [1,2,3,4,5,6,7,8,9]. An ever-growing list of candidate anti-aging actives have been studied and purported to reduce or even reverse skin aging [2,10]. However, few of these agents have strong evidence demonstrating that the anti-aging effects elucidated in cell culture, animal models, or cosmetic studies (which assess wrinkling and epidermal surface read-out parameters) [11] translate to a significant improvement in accepted molecular aging biomarkers in full-thickness human skin.

Therefore, one key challenge in the ongoing quest to identify key drivers of human skin rejuvenation [3,12] is to document that candidate anti-aging actives robustly improve core molecular and cellular aging biomarkers. Another challenge is to identify robustly effective actives that are well tolerated, widely available, economical, and that—most importantly—counteract human skin aging on multiple levels of pathogenesis, thereby increasing the likelihood of eliciting profound, long-lasting anti-aging effects.

One particularly attractive candidate is melatonin, a pleiotropic, evolutionarily highly conserved indoleamine neurohormone. Though best known for its sleep-enhancing properties via its actions on the central circadian clock, melatonin also unfolds potentially synergistic multi-level anti-aging effects [13,14,15,16]. Melatonin itself is a potent, highly diffusible antioxidant whose nocturnal peak and total serum level naturally decrease with age [17,18,19,20]. It increases the expression and activity of both ROS scavenging and DNA repair enzymes and may also positively interfere with epigenetic aging pathomechanisms [21,22,23,24,25]. It also prevents excessive apoptosis [26], including in human hair follicle epithelial stem cells ex vivo [27], and limits not only oxidative stress [3,26,28] and UV radiation (UVR)-induced damage [3,18,26,28,29,30] but also inflammation [31,32], and thus potentially even inflammaging [6].

Moreover, melatonin is well tolerated, inexpensive [33], and has good skin penetration, thus being recommended for topical application [34,35]. Its amphiphilic nature enables it to penetrate biological membranes better than any other known endogenous antioxidant and to thereby effectively protect DNA, proteins, and lipids against oxidative damage, especially in the nucleus and mitochondria [36,37]. This property is important because topical application prevents the problems of rapid hepatic degradation and off-target sedative effects of systemically administered melatonin and facilitates cutaneous accumulation in excess of the endogenous levels synthesized within human skin and its appendages [16,21,22,23,24,38,39].

Given that the key biological effects of melatonin are mediated by the activation of signaling through the G-protein-coupled membrane-bound receptors MT1 and MT2 [40,41,42] and that both receptors are expressed in human skin [43,44], skin is capable of responding to melatonin stimulation. Indeed, contrary to widely held beliefs in the literature [39], we have recently shown that longer-term stimulation of organ-cultured human eyelid skin with high-dose melatonin significantly increases epidermal melanin content, melanocyte dendricity and cell number as well as overall epidermal thickness [45]. This is also important from an aging research perspective, since loss of epidermal pigmentation and increasing epidermal atrophy are key features of aging skin, while the reversal of these morphological phenomena indicates skin rejuvenation [3]. Moreover, senescent epidermal melanocytes may even function as an important driver of human epidermal aging [46].

Despite the abundance of arguments presented here that recommend melatonin as an ideal cutaneous anti-aging active, it remains to be conclusively demonstrated that melatonin indeed significantly improves widely accepted core molecular biomarkers of human skin aging at the protein level in intact human skin. The current study critically probes this working hypothesis under serum-free, highly standardized ex vivo conditions using our well-established human skin organ culture assay [45,47] to assess whether melatonin administration significantly alters key read-out parameters of cellular senescence, epidermal mitochondrial function, and dermal extracellular matrix composition.

We purposely employed very thin, full-thickness, aged human eyelid skin [48,49,50] due to its chronic UV exposure and cosmetic relevance to both individuals and the skin anti-aging industry. In order to generate proof of principle, we administered melatonin to the culture medium, thus imitating a “systemic” mode of application and circumventing potential confounders arising from topical application, including transcutaneous penetration and intraepidermal melatonin metabolism.

Specifically, we asked whether the “systemic” administration of high-dose melatonin (100 µM and 200 µM) [45] significantly changes epidermal S6 phosphorylation (a direct marker of mTORC1 activity [51]), collagen 17A1 (the key stem-cell-niche-associated transmembrane matrix molecule [52,53,54]), Matrix Metalloproteinase 1 (MMP-1, the activity of which is associated with degradation of collagen I and collagen III fibers [55]). We also assessed the protein immunoreactivity of sirtuin-1 ((SIRT-1), which takes part in deacetylation of key substrates—histones or p53—involved in the regulation of cell metabolism and cell cycle [56]), of lamin-B1 (the main component of nuclear lamina stabilizing the nucleus [57,58]), p16^INK4^ (tumor suppressor promoting the senescence in cell cycle [59,60]), γH2A.x (phosphorylated histone protein H2A.x, a marker of double-strand breaks in DNA [61]), the mitochondrial markers TFAM, MTCO-1, and VDAC/Porin, since mitochondrial dysfunction is associated with aging [33,47,62,63], and the recently identified key driver of human skin rejuvenation, VEGF-A [3,17]. In the dermis, we analyzed protein expression of fibrillin-1 and collagen I, matrix proteins produced by fibroblasts, the reduced production and increased degradation of which produce key clinical features of skin aging like wrinkle formation and loss of elasticity [64,65].

## 2. Results

To address the above questions, upper eyelid skin fragments from 5–6 distinct, independent donors aged 49 to 77 years were organ-cultured in serum-free, supplemented medium for 3 or 6 days in the presence of melatonin (100 µM, 200 µM) or the vehicle (0.2% ethanol) only, and then cryoembedded in OCT, shock-frozen in liquid nitrogen, and immunostained with the corresponding primary antibodies and as explained in Section 4. Protein immunoreactivity results were assessed via quantitative immunohistomorphometry (qIHM) of carefully defined reference areas in non-consecutive cryosections of four sections per condition in two different skin fragments per analyzed biomarker for each donor, derived from 5–6 independent organ cultures. First, we investigated selected aging-related biomarkers in the epidermis (see Section 2.1, Section 2.2, Section 2.3, Section 2.4 and Section 2.5) and then in the dermis (see Section 2.6).

### 2.1. Melatonin Downregulates mTORC1 Activity in Aged Human Skin Ex Vivo

Cell and tissue aging is associated with increased mTORC1 activity, and the mTORC1 pathway significantly promotes the senescence-associated secretory phenotype of senescent cells [66]. To gauge mTORC1 activity via qIHM, we examined the expression of phosphorylated ribosomal protein S6 (p-S6), a sensitive direct downstream target of mTORC1 signaling, as an aging biomarker [51,67]. This showed a significant reduction in p-S6 protein expression within the epidermis, particularly notable in the stratum spinosum, after three days of culture with a concentration of 200 µM melatonin when compared to the vehicle control group (Figure 1a,b). This demonstrates that melatonin can significantly reduce aging/senescence-promoting mTORC1 activity within healthy, aged human skin ex vivo.

### 2.2. Melatonin Modulates Expression of MMP-1 and COL17A1 in Skin Tissue

MMP-1, also known as interstitial collagenase or collagenase 1, is mainly secreted by epidermal keratinocytes and dermal fibroblasts and breaks down fibrillar collagens type I and III [55]. Increased MMP-1 activity is critically involved in dermal aging [68,69], even though MMP-1 secretion mainly arises from the epidermal keratinocytes, e.g., in response to photodamage [47,70]. Therefore, it is interesting to note that MMP-1 protein levels significantly decreased across the entire epidermis following a 6-day incubation of eyelid skin with 200 µM melatonin. (Figure 1c,d). Though it remains to be evaluated, e.g., via in situ enzyme histochemistry [47], whether this translates into a change in intraepidermal or intradermal MMP-1 activity, this suggests that melatonin can downregulate excessive MMP-1 expression and thus collagen fibril degradation.

Collagen 17A1 (COL17A1), a structural component of hemidesmosomes mainly expressed by basal keratinocytes, has surfaced as a key stem-cell-niche-associated extracellular matrix protein whose progressive degradation during aging promotes epithelial stem cell differentiation and thus depletion of the stem cell niche [45,54]. This prompted us to also evaluate the expression of COL17A1 (which is decreased in UV-irradiated keratinocytes showing high levels of MMP-1 [71]). Our data showed a significant increase in COL17A1 expression in the epidermal basement membrane zone after 6 days of treatment with 200 µM melatonin (Figure 1e,f). Our findings suggest that melatonin contributes to the prevention of COL17A1 degradation, which is potentially associated with the activation of epidermal stem cells [72] directly or indirectly via a reduction in MMP-1 expression. This suggests that melatonin promotes skin rejuvenation or retards ex vivo-skin aging [xx] at these concentrations.

### 2.3. Melatonin Improves Mitochondrial Marker Expression

Mitochondrial dysfunction plays a crucial role in skin aging [33,63,73,74,75], leading us to also investigate the impact of melatonin on the expression of the key mitochondrial proteins cytochrome c oxidase I (MTCO-1) and mitochondrial transcription factor (TFAM), which control mitochondrial biogenesis and DNA synthesis [21,22,23]. We also examined voltage-dependent anion channel VDAC/Porin expression, as a useful screening marker for mitochondrial abundance. VDAC/Porin regulates both apoptosis and mitophagy and can control mitochondrial DNA (mtDNA) release during inflammatory response [44,45,46]. We found that 200 µM melatonin can significantly increase TFAM protein expression in the epidermis after 6 days of organ culture (Figure 2a,b), while MTCO-1 expression was significantly increased with 100 µM melatonin (Figure 2c,d). Epidermal VDAC/Porin protein expression was also significantly increased after 6 days of culture (Figure 2e,f). Since we had previously shown that MTCO-1, TFAM, and VDAC/Porin immunoreactivity correlate well with mitochondrial complex II/IV activity and mitochondrial biogenesis in human skin [47,62,76], these data strongly suggest that melatonin can improve aging-related mitochondrial dysfunction and number.

### 2.4. Melatonin Unfolds Differential Anti-Aging Activities in Human Skin: No Effect on Epidermal Lamin B1, Sirtuin-1, and Aging-Associated DNA Damage

Given the complex, multi-level pathobiology of human skin aging (see Section 1), we next assessed whether melatonin non-discriminatively improves additional core skin-aging associated biomarkers across the board. To this end, we examined p16^INK4^ protein expression, a key biomarker for senescent cells [28,32,60,77]; lamin B1, a nuclear lamina component whose abundance declines with age/senescence [36,78,79]; sirtuin-1 (SIRT-1), an NAD^+^-dependent deacetylase enzyme that is crucially involved in cell and tissue aging [80,81,82,83]; and γH2A.x formation resulting from histone H2A.x phosphorylation as a marker of DNA damage [24,61,84,85].

Interestingly, however, we did not observe any robust and significant changes in these core aging-associated biomarkers following melatonin supplementation (Appendix A). 

This shows that the multi-level anti-aging effects exerted by high-dose melatonin in human skin (Figure 1a–f, Figure 2a–f, Figure 3a,b and Figure 4a–d) are by no means present across the board, and instead are more differential and selective than the very broad range of anti-aging/senotherapeutic effects of melatonin reported in the cell culture and animal [86,87,88,89] had led us to expect.

### 2.5. Melatonin also Promotes Intraepidermal Production of VEGF-A, the Key Driver of Skin Rejuvenation, Ex Vivo

We recently showed that the key angiogenesis regulatory growth factor, vascular endothelial growth factor A (VEGF-A), is a critical driver of human skin rejuvenation both in vivo and ex vivo [3]. In line with the “angiogenesis hypothesis of aging”, which proposes that a reduction in the production of angiogenic factors and subsequently decreased capillary density promote tissue aging [90], we were interested in checking whether melatonin exerted any impact on epidermal VEGF-A protein expression. Melatonin has mainly been discussed in the literature as a candidate anti-angiogenesis factor that may counteract pro-angiogenic VEGF-A effects [91,92,93]. However, it has conversely also been reported to ameliorate aging-associated impaired angiogenesis [94] and to promote VEGF and/or VEGF receptor expression under some conditions [95,96]. Unexpectedly, qIHM showed that melatonin treatment significantly increased the epidermal protein expression level of VEGF-A (Figure 3a,b). This could represent an indirect mechanism by which melatonin might promote human skin rejuvenation [3].

### 2.6. Melatonin Increases Dermal Fibrillin-1 and Collagen I Content

Having assessed these skin-aging-associated biomarkers primarily in the epidermis, we finally turned to the dermis. Increased MMP-1 activity during skin aging results in a reduction in fibrillin-rich microfibrils, which play a crucial role in maintaining skin elasticity and integrity [97,98]. We therefore assessed dermal fibrillin-1, which decreases in abundance and displays a deterioration in orderly spatial arrangement with aging [99,100,101]. qIHM revealed a significant increase in fibrillin-1 expression, with maximal effects reached with only 100 µM melatonin applied for 3 days, indicating that even lower doses and treatment durations yield improved fibrillin-1 abundance (Figure 4a,b). Qualitative analysis reaffirmed these impressive improvements in intradermal fibrillin-1, with the melatonin-treated group also displaying improved orderly microfibril arrangement (Figure 4b).

Finally, we investigated melatonin’s impact on the intradermal expression of collagen I protein, the abundance of which declines during skin aging [53,102,103,104]. qIHM revealed an increase (*p* = 0.0688) in the protein expression of collagen type I after 6 days of treatment with 200 µM melatonin (Figure 4c,d).

## 3. Discussion

This ex vivo study provides the first conclusive immunohistological evidence that melatonin indeed exerts long-suspected anti-aging effects on multiple levels in human eyelid skin—both in the epidermis and the dermis. Namely, “systemic” high-dose melatonin significantly reduced mTORC1 activity and MMP1 expression, while COL17A1 protein expression was significantly increased and both fibrillin-1 expression and fiber structure were improved. Unexpectedly, melatonin also increased the protein expression of VEGF-A, which was recently discovered to be a key driver of human skin rejuvenation [3]. Yet, melatonin’s anti-aging effects in our assay were also surprisingly differential since three other core aging biomarkers (SIRT1, p16^INK4^, lamin B1) remained unaffected (Table 1).

The changes observed in our study may look relatively small if compared to cell culture data in vitro. However, one needs to take into account that these reflect 10–20% changes within a whole human organ that is constituted of multiple different cell types after a relatively short 6-day treatment period. Thus, it is reasonable to expect that the subtle, but significant, melatonin effects observed during such a short treatment window would be maximized upon long-term treatment during a clinical melatonin study, e.g., over several months. Our study was designed to provide proof of concept of whether high-dose melatonin can indeed exert anti-aging effects in intact human skin. A “systemic” mode of administration, i.e., melatonin addition to the medium, was chosen so as to guarantee optimal melatonin access to the tissue. If our findings were to be translated into a clinical use of melatonin for anti-skin aging purposes, the most logical mode of application would be a topical one so that high-dose melatonin, which is highly amphiphilic and thus has excellent cell and tissue penetration capacity through human skin [34,35], can be administered directly to the target organ without massive loss of activity due to liver metabolism.

That melatonin unfolded these differential anti-aging effects in aged human eyelid skin is particularly intriguing and clinically relevant, since this skin region is highly susceptible to photoaging and is a major target of commercial anti-aging products. One limitation of this model is that these skin samples are very difficult to source, which made it impossible to test more than two concentrations of melatonin and restricted us to two observational time points during organ culture. Moreover, while it may be desirable to include a positive control active, the very limited availability of eyelid skin did not permit such a control to be added. However, melatonin itself has been amply documented in numerous in vitro and in vivo studies in many different species and test systems to exert profound anti-aging properties [105,106,107,108,109,110,111]—but not yet in human skin ex vivo. As such, melatonin itself can arguably be considered to serve as a “positive control”.

Mitochondrial dysfunction is a major contributor to aging [112,113,114], and melatonin has previously been reported to improve mitochondrial function on multiple levels in various cells and systems [107,115,116,117]. In line with this, 200 µM melatonin increased TFAM and VDAC/Porin protein expression levels in aged human skin. This is particularly interesting since it has been recently shown that TFAM is significantly reduced in elderly people, while VDAC/Porin is also decreased but not significantly [112].

Previous longevity studies have underscored the importance of SIRT1 as an aging biomarker and have demonstrated that its expression decreases in in vitro aging models [118], while SIRT-1 transcription in aging murine ovaries increases after melatonin treatment [119]. However, at the protein level, we did not observe a melatonin-induced change in SIRT-1 expression, which is similar to what we previously observed with other skin anti-aging molecules, triiodothyronine and thyroxine [47]. The same lack of significant effects was noted for the key senescence marker p16^INK4^ [32], which increases under UVB-induced stress in epidermal keratinocytes [120]. The latter also shows loss of lamin B1 expression [14,120]. Yet again, melatonin did not significantly alter the level of lamin B1 protein expression ex vivo. Even though melatonin has been documented to promote DNA repair [16,39,121,122,123], no reparative effect on aging-associated DNA damage, as assessed via qIHM of γH2A.x histone phosphorylation [24,84,85], was seen in our human eyelid organ culture for the duration of the assay (3 and 6 days of culture).

Skin aging also results in key changes in the composition of the dermal extracellular matrix, contributing to the clinical appearance of wrinkles and reduced elasticity. Total collagen quantity decreases in response to increased MMP-1 activity and ROS-induced TGF-β signaling impairment [124,125]. The reduction in MMP-1 coupled with melatonin’s known antioxidant properties may well have contributed to the observed relative increase in the steady-state level of detectable dermal collagen I (note that MMP-1 is hypothesized to be expressed primarily in the epidermis before diffusing into the papillary dermis [47]). Moreover, even though the observed slight increase in collagen I expression was not significant, this suggests that melatonin may maintain skin tensile strength, given collagen I’s known functional as a structural scaffold.

Furthermore, COL17A1 is closely linked to skin aging, as its levels significantly decrease with age [53]. Researchers have focused on developing significant peptide derivatives and innovative techniques such as radiofrequency and ultrasound to enhance COL17A1 levels in the skin, leading to improvements in facial wrinkles [72]. The reduction in COL17A1 levels as we age is associated with the deterioration of hemidesmosomes, resulting in the microdelamination of basal cells in the skin—an essential factor in skin aging [124,126]. COL17A1’s role in skin aging is apparent in various skin stem cells, including the epidermal stem cells, where it regulates cell competition, self-renewal capacity, and stem cell maintenance by controlling the balance of symmetric and asymmetric cell divisions. The disruption in the COL17A1 balance eventually causes age-related epidermal atrophy and fragility [72]. Our findings regarding melatonin’s positive regulation of COL17A1 suggest its potential as a therapy for skin aging and protecting epidermal stem cells, which are important for skin self-renewal.

In theory, higher concentrations of melatonin might have exerted even more anti-aging benefits than we report here. Though MT1- and MT2-mediated signaling eventually reach a receptor saturation maximum, beneficial MT1/2-independent effects of melatonin, including signaling via cytosolic MT3 or a long-discussed but elusive nuclear melatonin receptor, as well as direct ROS scavenging and enzyme-activity-modulatory effects of melatonin [127,128,129,130,131], could well be exerted by high-dose topical melatonin.

Though “systemically” administered high-dose melatonin leads to substantial anti-aging effects in human eyelid skin organ culture, these effects do not seem to be quite as universal in human skin as has previously been discussed [15,132,133], at least under our assay conditions and with the biomarkers that were assessed. The major open question in this context is what exactly determines melatonin’s apparent selectivity for some but not other aging-related pathways in human eyelid skin. While our organ culture system is a very challenging tool for in-depth mechanistic research that addresses this critical question, the aging-related biomarkers melatonin impacted significantly at least provide important leads as to selected key pathways that deserve special attention; these include mTORC1-dependent signaling, the enzymatic controls of collagens I and COL17A1 and elastic fiber degradation and synthesis, and selected core controls of mitochondrial DNA and function (TFAM, VDAC/porin). Also, it would be interesting to systematically compare the selective and differential anti-aging effects exerted by melatonin in this new eyelid skin organ culture assay with those exerted by actives whose significant anti-aging properties we have most recently documented in human truncal skin ex vivo, namely VEGF-A [1], caffeine, the natural pheromone, 2,5-dimethylpyrazine, and tretinoin [134].

Arguably the most surprising effect of melatonin on aged human skin elucidated in this study is its stimulation of intraepidermal protein expression of VEGF-A, an astoundingly potent driver of human skin rejuvenation on multiple levels of skin aging, both in vivo and ex vivo [3]. Though much of this effect appears to be mediated by enhanced angiogenesis, VEGF-A also reduces oxidative stress and improves oxidative damage repair [3]. Our data raise the intriguing possibility that melatonin recruits endogenous, intracutaneously generated VEGF-A to potentiate its own anti-aging properties within human skin. Of course, the merely correlative evidence presented here needs to be followed up with VEGF-A-neutralizing antibody and anti-VEGF (e.g., bevacizumab) studies in order to probe whether this upregulation of VEGF-A and the modulation of angiogenesis are functionally important for mediating any of the anti-aging effects of melatonin shown in Table 1. Interestingly, VEGF-A itself significantly reduces p16^INK4^ and improves SIRT-1 expression in aged human epidermis in vivo [3], while the melatonin doses tested here did not do so (Appendix A). Therefore, it is reasonable to speculate that the differential melatonin-induced anti-aging effects reported here are not primarily driven by increased VEGF-A.

Finally, while a convincing clinical study is undoubtedly needed to investigate the anti-aging properties of melatonin in human skin, better preclinical justification for designing and executing such a study in human subjects is required. Our preclinical study provides the long-missing preclinical evidence—directly in the human target organ itself—that melatonin indeed unfolds significant and differential anti-aging activities in human skin ex vivo. Since at least one registered clinical anti-aging trial is investigating the impact of melatonin on cognition and brain health (clinicaltrials.gov, identifier: NCT03954899), we hope that the current preclinical study in the human target organ will also encourage the initiation of clinical trials of melatonin on human skin aging.

## 4. Materials and Methods

We used upper eyelid skin (given that it is sun-exposed and targeted by anti-aging cosmetics) from 5–6 donors (49–77 years old; mean: 61.33 years) undergoing blepharoplasty [135,136,137,138,139,140]. This age range was selected since this corresponds to the main target population for anti-aging cosmetic products and melatonin serum levels decline with age [19,141], which may render aged skin relatively deficient in melatonin signaling and potentially more responsive to exogenous melatonin stimulation.

The use of anonymized, discarded human tissue was considered to represent non-human research and exempted under 45 CFR46.101.2 by the IRB of the University of Miami Miller School of Medicine (Miami, FL, USA).

### 4.1. Eyelid Skin Organ Culture 

Serum-free organ culture began on the day of surgery and proceeded according to procedure described earlier [45]. Briefly, 4 mm skin punch biopsies were taken from the surgical skin samples and placed in culture in supplemented serum-free William’s E medium. After 24 h of equilibration, skin biopsies were assigned randomly to 3 experimental study conditions: vehicle (control, 0.2% ethanol), 100 µM melatonin, and 200 µM melatonin. We have previously demonstrated that such “supraphysiological” doses of melatonin result in increased epidermal melanocyte and keratinocyte proliferation in human eyelid skin ex vivo, suggesting that they may promote skin rejuvenation [45]. We wished to exactly reproduce the organ culture conditions in which we demonstrated these remarkable epidermal pigmentation and thickness benefits [45]. These melatonin doses were originally chosen to mimic the high concentrations that can be achieved with cosmeceutical application and to avoid false-negative results, given that available literature indicates high concentrations of melatonin may be necessary to achieve phenotypic effects [39,142,143,144,145,146,147,148]. Furthermore, the actual intracutaneous levels of endogenously produced melatonin are unknown but are likely significantly higher than serum levels, so these “supraphysiological” doses may even be, in fact, physiologic [16,149,150].

Culture was run for 3 or 6 days, and the treatment was repeated every other day. Afterwards, the skin punches were embedded in OCT (Thermo Scientific, Waltham, MA, USA), snap-frozen in liquid nitrogen, and cut into 7 μm cryosections using a Cryostar NX50 cryostat (Thermo Scientific). The scarcity of eyelid samples available for testing prevented dose–response assays from being conducted. Instead, only two time points and two relatively high doses could be tested, as in our previous work [45]. The addition of melatonin to culture medium imitates systemic application.

### 4.2. Immunohistochemical Staining 

Following fixation, cryosections were pre-incubated at room temperature (RT) for 30 min, followed by overnight incubation at 4 °C with the primary antibody. The following primary antibodies were used: rabbit monoclonal anti-phospho-S6 ribosomal protein (Ser235/236) (1:200; Cell Signaling, Danvers, MA, USA, #4858), rabbit monoclonal anti-COL17A1 [EPR14758] (1:200; Abcam, Cambridge, MA, USA, #ab186415), mouse monoclonal anti-MMP1 (1:100; Biolegend, San Diego, CA, USA, #634702), rabbit monoclonal anti-TFAM [EPR12285](1:500; Abcam, #ab176558), rabbit monoclonal anti-VDAC1/Porin (1:100; Abcam, #ab15895), rabbit monoclonal anti-MTCO1 [EPR19628] (1:50; Abcam, #ab203912), rabbit monoclonal anti-human Alexa Fluor^®^ 488 Anti-VEGFA [EP1176Y] (1:500, Abcam, #ab206886), rabbit monoclonal anti-SIRT1 [E104] (1:200; Abcam, #ab32441), rabbit monoclonal anti-lamin B1 [EPR8985(B)] (1:100; Abcam, #ab194109), rabbit monoclonal anti-phospho-histone γH2A.X (Ser139) (1:1500; Cell Signaling, #2577S), rabbit monoclonal anti-CDKN2A/p16^INK4a^ [EPR1473] (1:250; Abcam, #ab108349), mouse monoclonal anti-fibrillin-1 biotinylated [11C1.3] (1:800; Abcam, #ab24826), and mouse monoclonal anti-Collagen I(1:500; Novus Bio, Centennial, CO, USA, #NB600-450).

All antibodies were diluted with PBS, TBS, or TNT as appropriate. After 3 consecutive 5 min washes, slides were then incubated with the appropriate fluorescent-labeled secondary antibody for 45 min at RT. All slides were embedded using DAPI/Fluoromount (Southern Biotechnologies, Birmingham, AL, USA) and stored at −20 °C. The detailed description of immunohistochemical staining is provided in Table 2.

### 4.3. Image Analysis 

Images were taken using the BZ-X800 All-in-one Fluorescence Microscope (Keyence Corporation, Itasca, IL, USA) with magnification at 200 X and the exposure time kept constant for all images taken of a particular stain and donor. This approach is consistent with previous studies [151,152]. Analysis was then conducted using ImageJ (NIH). p16^INK4^ and γH2A.x were measured by counting the proportion of cells in the lowest two cells lines of the epidermis that were double-positive for both DAPI and the marker of interest. All other epidermal markers (MTCO-1, SIRT-1, TFAM, VDAC/Porin, MMP1, VEGF-A, p-S6, lamin-B1) were assessed according to the intensity level of fluorescence in the defined reference area (the full thickness of the epidermis, excluding stratum corneum). Dermal markers fibrillin-1 and collagen I were analyzed according to the intensity level of fluorescence in the first 200 µm subepidermal dermis, corresponding to the papillary dermis, while COL17A1 in the first 100 µm. All results were then normalized to the vehicle.

### 4.4. Statistical Analysis 

All data are shown either as the fold change of the mean ± SEM or as mean ± SEM. First, normality and lognormality tests (namely D’Agostino and Pearson, Anderson–Darling, Shapiro–Wilk, Kolmogorov–Smirnov) were performed, and then analysis was performed using Student’s *t*-test (for Gaussian distribution) or Mann–Whitney U-test (non-Gaussian distribution) (Graph Pad Prism 9, GraphPad Software, La Jolla, CA, USA). A *p* < 0.05 was considered significant.

## Figures and Tables

**Figure 1 ijms-24-15963-f001:**
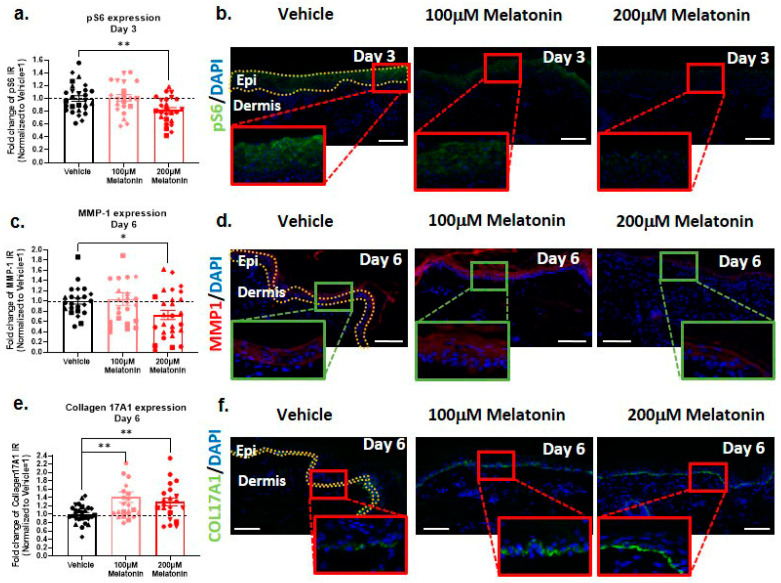
Melatonin reduces aging-associated mTORC1 activity and MMP-1 protein expression and promotes COL17A1 protein expression in aged human epidermis ex vivo. (**a**,**b**) We found 200 µM melatonin significantly reduced p-S6 expression in epidermis on day 3. Quantitative immunohistomorphometry and representative images of p-S6 expression, showing the areas of evaluation in the control group. Mean ± SEM; n = 21–28 non-consecutive skin sections from 6 donors; Mann–Whitney test; ** *p* < 0.01. (**c**,**d**) We found 200 µM melatonin significantly decreased MMP-1 expression in epidermis on day 6. Quantitative immunohistomorphometry graph and representative images displaying differences in MMP-1 intensity expression between treatment conditions on day 6. Mean ± SEM; n = 23–27 non-consecutive skin sections from 6 donors; Mann–Whitney test; * *p* < 0.05. (**e**,**f**) Concentration of 100 µM and 200 µM melatonin increased COL17A1 expression in the basement membrane zone on day 6. Quantitative immunohistomorphometry and representative images for COL17A1 expression. Mean ± SEM; n = 22–27 non-consecutive skin sections from 6 donors; Mann–Whitney test; ** *p* < 0.01. Epi: epidermis. Scale bar: 100 µm.

**Figure 2 ijms-24-15963-f002:**
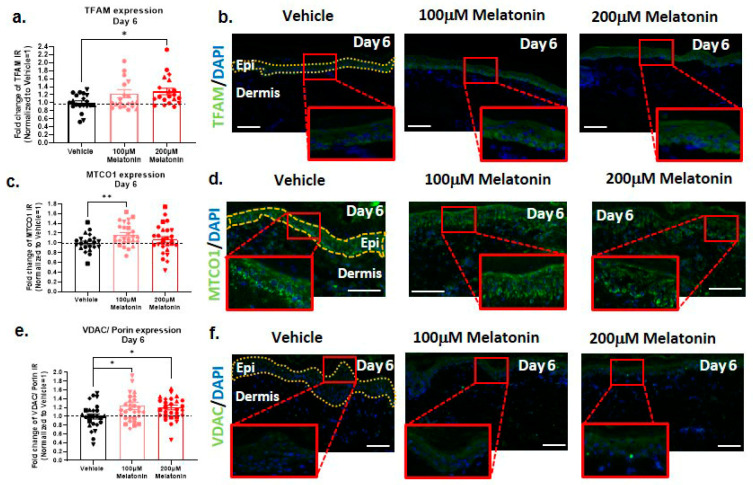
Melatonin significantly upregulates mitochondrial activity and increases mitochondrial biogenesis in human epidermis ex vivo. (**a**,**b**) We found that 200 µM melatonin significantly upregulated TFAM expression in epidermis on day 6. Quantitative immunohistomorphometry and representative images of TFAM expression. Mean ± SEM; n = 20–21 non-consecutive skin sections from 5 donors; Mann–Whitney test; * *p* < 0.05. (**c**,**d**) Treatment with 100 µM melatonin showed upregulation of MTCO-1 in epidermis on day 6. Quantitative immunohistomorphometry and representative pictures of MTCO-1 expression. Mean ± SEM; n = 22–27 non-consecutive skin sections from 6 donors; Mann–Whitney test; ** *p* < 0.01. (**e**,**f**) Treatment with 100 µM and 200 µM melatonin significantly increased VDAC/Porin expression on day 6. Quantitative immunohistomorphometry and representative images of VDAC expression. Mean ± SEM; n = 25–30 non-consecutive skin sections from 6 donors; Mann–Whitney test; * *p* < 0.05. Epi: epidermis. Scale bar: 100 µm.

**Figure 3 ijms-24-15963-f003:**
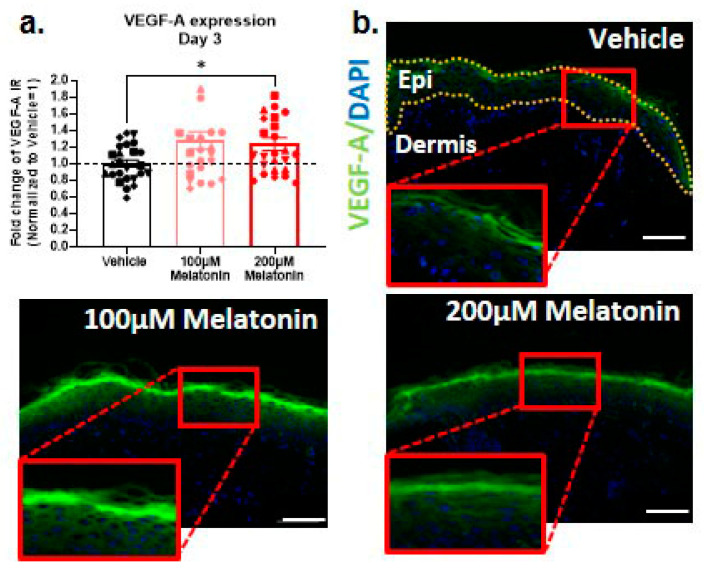
Melatonin significantly stimulates the skin anti-aging factor VEGF-A. (**a**,**b**) We found that 200 µM melatonin significantly stimulates the skin anti-aging factor VEGF-A in epidermis on day 3. Quantitative immunohistomorphometry and representative images. Mean ± SEM; n = 24–25 non-consecutive skin sections from 6 donors; Mann–Whitney test; * *p* < 0.05 Epi: epidermis. Scale bar: 100 µm.

**Figure 4 ijms-24-15963-f004:**
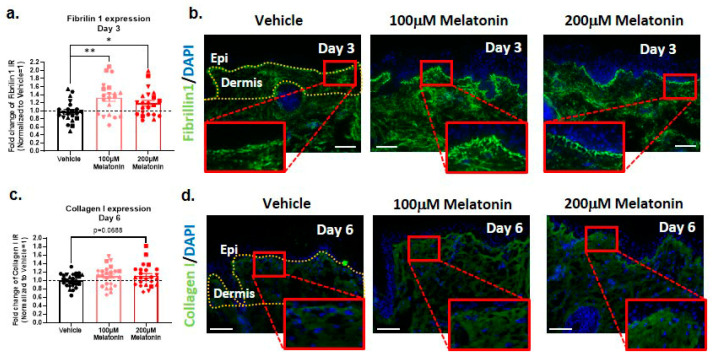
Melatonin significantly increased the expression of fibrillin I and collagen I in the dermis, indicating reduced dermal aging. (**a**,**b**) Amounts of 100 µM and 200 µM melatonin significantly upregulated fibrilin-1 expression in dermis on day 3. Quantitative immunohistomorphometry and representative images of fibrilin-1 expression. Qualitative improvement in the orderly arrangement of fibrillin 1^+^ microfibrils after treatment with 100 µM and 200 µM melatonin can also be seen. Mean ± SEM; n = 20–22 non-consecutive skin sections from 6 donors; Student’s *t*-test, * *p* < 0.05, ** *p* < 0.01. (**c**,**d**) Expression of collagen I in dermis on day 6 increased after melatonin treatment. Quantitative immunohistomorphometry and representative images of collagen I expression. Mean ± SEM; n = 24–28 non-consecutive skin sections from 6 donors; Student’s *t*-test; non-statistically significant. Epi: epidermis. Scale bar: 100 µm.

**Table 1 ijms-24-15963-t001:** Overview of the impact of melatonin administration on key skin aging biomarkers. ↑ = increased, ↔ = no change, ↓ = decreased, compared to the vehicle. Mann–Whitney or Student’s *t*-test, * *p* < 0.5, ** *p* < 0.01.

Marker	With Melatonin, Day 3	With Melatonin, Day 6
	100 µM	200 µM	100 µM	200 µM
p-S6		** 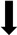		
MMP-1				* 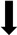
COL17A1			** 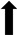	** 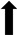
TFAM				* 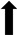
VDAC/Porin	* 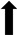		* 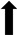	* 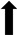
MTCO1			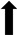	
VEGF-A	* 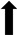		* 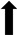	
Fibrillin 1	** 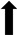	* 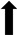		
Collagen I				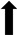
p16^INK4^				
Lamin B1				
SIRT1				
γH2A.x				

**Table 2 ijms-24-15963-t002:** List of antibodies and immunohistochemical staining treatments used in the study.

	Antigen	Fixation	Blocking and/or Permeabilization	Primary Antibody	Secondary Antibody
Epidermis	p-S6	4% PFA, 10 min at RT	N/A	rabbit monoclonal anti-phospho-S6 ribosomal protein (Ser235/236) (1:200; Cell Signaling, #4858)	goat monoclonal anti-rabbit IgG-Alexa Fluor 488 (1:200; Invitrogen, Waltham, MA, USA, A11034)
	COL17A1	Acetone, 10 min at −20 °C	N/A	rabbit monoclonal anti-Collagen 17A1[EPR14758] (1:200; Abcam, #ab186415)	goat monoclonal anti-rabbit IgG-Alexa Fluor 488 (1:400; Invitrogen, A11034)
	MMP-1	4% PFA, 10 min at RT	10% goat serum in PBS	mouse monoclonal anti-MMP1 (1:100; Biolegend, #634702)	goat monoclonal anti-mouse IgG-Alexa Fluor 555 (1:200; Invitrogen, A32727)
	TFAM	Acetone, 10 min at −20 °C	0.1% Triton X-100 in PBS	rabbit monoclonal anti-mtFA (TFAM) [EPR12285] (1:500; Abcam, #ab176558)	goat monoclonal anti-rabbit IgG-Alexa Fluor 488 (1:400; Invitrogen, A11034)
	VDAC1/Porin	Methanol, 10 min at −20 °C	10% goat serum and 0.3% Triton X-100 in PBS	rabbit monoclonal anti-VDAC1/Porin (1:100; Abcam, #ab15895)	goat monoclonal anti-rabbit IgG-FITC (1:200; Jackson Immuno Research, West Grove, PA, USA, 111-095-144) followed by an amplification with goat monoclonal anti-FITC-Alexa Fluor^®^ 488 (1:700; Invitrogen, A11096)
	MTCO-1	4% PFA, 10 min at RT	10% goat serum and 0.3% Triton X-100 in PBS	rabbit monoclonal anti-MTCO1 [EPR19628] (1:50; Abcam, #ab203912)	goat monoclonal anti-rabbit IgG-FITC (1:200; Jackson Immuno Research, 111-095-144) followed by an amplification with goat monoclonal anti-FITC-Alexa Fluor^®^ 488 (1:700; Invitrogen, A11096)
	VEGF-A	4% PFA, 10 min at RT	N/A	rabbit monoclonal anti-human Alexa Fluor^®^ 488 Anti-VEGFA [EP1176Y] (1:500, Abcam, #ab206886)	N/A
	SIRT-1	Acetone, 10 min at −20 °C	10% goat serum and 0.3% Triton X-100 in PBS	rabbit monoclonal anti-SIRT1 [E104] (1:200; Abcam, #ab32441)	goat monoclonal anti-rabbit IgG-FITC (1:200; Jackson Immuno Research, 111-095-144) followed by an amplification with goat monoclonal anti-FITC-Alexa Fluor^®^ 488 (1:700; Invitrogen, A11096)
	Lamin B1	4% PFA, 10 min at RT	N/A	rabbit monoclonal anti-Lamin B1 [EPR8985(B)] (1:100; Abcam, #ab194109)	goat monoclonal anti-rabbit IgG-Alexa Fluor 488 (1:400; Invitrogen, A11034)
	γH2A.x	4% PFA, 10 min at RT	10% goat serum in PBS	rabbit monoclonal anti-phospho-histone H2A.X (Ser139) (γH2A.X) (1:1500; Cell Signaling, #2577S)	goat monoclonal anti-rabbit IgG-Alexa Fluor 555 (1:400; Invitrogen, A21428)
	p16^INK4^	4% PFA, 10 min at RT	10% goat serum, 0.1% Triton-X100, and 0.2% Saponin in TBS	rabbit monoclonal anti-CDKN2A/p16^INK4a^ [EPR1473] (1:250; Abcam, #ab108349)	goat monoclonal anti-rabbit IgG-FITC (1:200; Jackson Immuno Research, 111-095-144) followed by an amplification with goat monoclonal anti-FITC-Alexa Fluor^®^ 488 (1:700; Invitrogen, A11096)
Dermis	Fibrillin-1	Acetone, 10 min at −20 °C	3% H_2_0_2_ in TBS, followed by pre-treatment with Avidin and Biotin and then 10% goat serum in TNT	mouse monoclonal anti-Fibrillin1 biotinylated [11C1.3] (1:800; Abcam, #ab24826)	amplification with FITC-conjugated tyramide
	Collagen I α-1	Acetone, 10 min at −20 °C	10% goat serum in PBS	mouse monoclonal anti-Collagen I alpha-1 (1:500; Novus Bio, #NB600-450)	goat monoclonal anti-mouse IgG-Alexa Fluor 488 (1:400; Invitrogen, A11001)

N/A—not applicable.

## Data Availability

The datasets used and/or analyzed during the current study are available from the corresponding authors on reasonable request.

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
