# Peer review of "Melatonin Exerts Prominent, Differential Epidermal and Dermal Anti-Aging Properties in Aged Human Eyelid Skin Ex Vivo"

_ijms, 2023, doi:10.3390/ijms242115963_

Round 1

Reviewer 1 Report

Comments and Suggestions for Authors

Accepted 

Comments on the Quality of English Language

Accepted with minor corrections

Author Response

Thank you.

Reviewer 2 Report

Comments and Suggestions for Authors

The study provides valuable insights into the potential anti-aging effects of melatonin on human eyelid skin. The findings reveal differential effects of melatonin on various skin aging biomarkers, shedding light on its selectivity in targeting specific pathways. Despite some limitations in terms of sample availability and dosing, this study contributes significantly to our knowledge of melatonin's potential as an anti-aging therapy for human skin, emphasizing its clinical relevance. However, I have some comments and concerns, especially concerning the methodologies employed in the study.

Comments:
The study mentions using upper eyelid skin from 3-4 donors, which is a relatively small sample size. It's important to consider whether this sample size provides sufficient statistical power for meaningful conclusions. In addition to the small sample size, the age range of donors, this span from 49 to 67 years, raises concerns. A narrower age range or age-matching could enhance the study's control over potential age-related variables. The authors should consider increasing the sample size and utilizing participants from a more homogeneous age group or provide a clearer rationale for why this is not feasible.

It is possible that Table 1 should also include data for a positive control - a substance known to affect skin aging markers. It is also noteworthy that doubling the melatonin dose from 100 to 200 µM has a significant impact on skin aging biomarkers. It is logical to ask what the minimum effective dose of melatonin is, at which its effects become evident. Additionally, it would be valuable to investigate how the effects change with higher melatonin dosages, such as 500 or 1000 µM. The rationale behind choosing 100 µM and 200 µM melatonin doses should be elaborated. The choice of these specific doses and the potential implications of their high concentrations should be discussed in greater detail.

Author Response

Reviewer 2:

The study provides valuable insights into the potential anti-aging effects of melatonin on human eyelid skin. The findings reveal differential effects of melatonin on various skin aging biomarkers, shedding light on its selectivity in targeting specific pathways. Despite some limitations in terms of sample availability and dosing, this study contributes significantly to our knowledge of melatonin's potential as an anti-aging therapy for human skin, emphasizing its clinical relevance. However, I have some comments and concerns, especially concerning the methodologies employed in the study.

Comments:
The study mentions using upper eyelid skin from 3-4 donors, which is a relatively small sample size. It's important to consider whether this sample size provides sufficient statistical power for meaningful conclusions. In addition to the small sample size, the age range of donors, this span from 49 to 67 years, raises concerns. A narrower age range or age-matching could enhance the study's control over potential age-related variables. The authors should consider increasing the sample size and utilizing participants from a more homogeneous age group or provide a clearer rationale for why this is not feasible.

Please note that we have previously published meaningful human skin organ culture data that have translated well into the clinic, using just 3 donors [1–6]. Nevertheless, as requested by the reviewer, we have now added 2 additional donors, aged 70 and 77 years - despite the very limited availability of human eyelid skin samples. These substantial new data fully support our previous conclusions and have been integrated into the thoroughly revised figures.

Concerning the rationale of the investigated age-range, we have selected an age range of 49-77 years since this corresponds to the main target population for anti-aging cosmetic products. Moreover, the rarity of eyelid skin becoming available from even older donors would essentially preclude to run a meaningful study. This is now explained in the revised Introduction and M+M section, p.11.

It is possible that Table 1 should also include data for a positive control - a substance known to affect skin aging markers. It is also noteworthy that doubling the melatonin dose from 100 to 200 µM has a significant impact on skin aging biomarkers. It is logical to ask what the minimum effective dose of melatonin is, at which its effects become evident.

While it may be desirable to include a positive control active, the very limited availability of eyelid skin did not permit to add such a control. Moreover, to be able to compare the effect of a positive control/known skin anti-aging molecule, we would need to test it in the same donor(s) we have perform our melatonin study. However, melatonin itself has been amply documented in numerous in vitro and in vivo studies in many different species and test systems to exert profound anti-aging properties [7–13] - but not yet in human skin ex vivo. As such, melatonin itself can arguably be considered to serve as a ‘positive control’. We have now explained this better in the revised Discussion, along with supporting references p.9.

Moreover, we now also explain in the revised Discussion - see p.10 - that it will be interesting to systematically compare in the new eyelid skin organ culture model the anti-aging effects of melatonin with that of other actives, whose significant anti-aging properties we have most recently documented in human truncal skin ex vivo, namely  VEGF-A [14], caffeine, the natural pheromone, 2,5.-dimethylpyrazine, and tretinoin [“Speed-aging” of human skin in serum-free organ culture ex vivo: An instructive novel assay for preclinical human skin aging research demonstrates senolytic effects of caffeine and 2,5-dimethylpyrazine Paus et al. Exp Dermatol, 2023 in press.]  

Additionally, it would be valuable to investigate how the effects change with higher melatonin dosages, such as 500 or 1000 µM. The rationale behind choosing 100 µM and 200 µM melatonin doses should be elaborated. The choice of these specific doses and the potential implications of their high concentrations should be discussed in greater detail.

Thank you for this excellent suggestion. The reviewer raises an important question about the minimum effective dosage. As expanded in section 4.1, we chose the concentrations of 100 and 200 µM based on our work in Sevilla et al. 2023 [15], where we demonstrated that these doses induced a remarkable increase in epidermal thickness and pigmentation while a lower concentration of melatonin (50 µM) did not demonstrate such effects. Other available research (see references [16,17], as well as [18]) also employs similarly high dosages, as human skin itself is a prominent site of extra-pituitary melatonin synthesis [19–21]. Moreover, high melatonin levels could readily be achieved by topical application: for example, a 0.1% melatonin face serum, corresponding to 43mM melatonin, has been used [22].

Since the physiological level of melatonin in human skin remains unknown, it is unclear whether our “supraphysiological” levels are actually higher than the physiological melatonin levels. However, it is known that melatonin serum levels decrease with age [23–25]. Thus, it is conceivable that aged human skin is characterized by a relative deficiency of melatonin levels compared to younger skin and may thus be particularly sensitive to melatonin stimulation.

The plausible and intriguing question whether higher concentrations of melatonin (e.g., 500-1000 µM) would exert even more anti-aging benefits deserves careful future study. Though one would presume that MT1- and MT2-mediated signaling would eventually reach a receptor saturation maximum, beneficial MT1/2-independent effects of melatonin, including signaling via cytosolic MT3 or a long-discussed nuclear melatonin receptor, as well as direct ROS scavenging and enzyme activity-modulatory effects of melatonin [26,27],  could well be  exerted by high-dose topical melatonin. We have now delineated this in the revised Discussion – p.10 -, supported by additional citations from the relevant published literature.

Reviewer 3 Report

Comments and Suggestions for Authors

This paper describes the molecular changes in cultured skin after media treatment with melatonin.  Melatonin is a well studied hormone in anti-aging and this study adds a little more molecular information.  What is needed is a convincing clinical study of melatonin in anti-aging.

The results for gene induction contradict other reports, e.g. SIRT1 has been reported to be induced (Ma et al., Rad Med Port 2:33-37, 2021).  Since each experimental method in these studies is different it calls into question the importance of each of these changes.

The changes are relatively small 10-20% in Fig 1 & 2.  Can these really result in significant changes in skin appearance?

The statistical analysis uses Mann-Whitney, which assumes each measure is independent.  However, the data is 15 samples drawn from only 3-4 cultures, so each data point is not independent. There are other methods to analyze such data and they should be used.

The authors should note that the doses used in this study are unrealistically high.  Melatonin has a short half-life of 1-2 hrs in vivo and is cleared by the liver.  Even high-dose oral use results in a few hours of serum levels that are 10 to 100-times lower than the doses used here.

Author Response

Reviewer 3:

This paper describes the molecular changes in cultured skin after media treatment with melatonin.  Melatonin is a well-studied hormone in anti-aging and this study adds a little more molecular information.  What is needed is a convincing clinical study of melatonin in anti-aging.

Thank you for this comment. While a convincing clinical study of the anti-aging properties of melatonin in human skin is undoubtedly needed, better preclinical justification for designing and executing such a study in human subjects is clearly needed. That is exactly what our study provides: the long-missing preclinical evidence - directly in the human target organ itself – that melatonin indeed unfolds significant and differential anti-aging activities in human skin.

We are aware of one registered clinical anti-aging trial with melatonin in the context of cognition and brain health (clinicaltrials.gov, identifier: NCT03954899). We hope that publication of our human skin organ culture study will encourages also initiation of a clinical trial of melatonin on human skin aging. These considerations have now been integrated into the revised Discussion, p.10.

The results for gene induction contradict other reports, e.g. SIRT1 has been reported to be induced (Ma et al., Rad Med Port 2:33-37, 2021).  Since each experimental method in these studies is different it calls into question the importance of each of these changes.

On this point, we respectfully disagree with the expert referee. If SIRT1 transcription is increased by melatonin in a very distinct assay system than the human skin organ culture assay used here, this by no means questions the validity of the – human protein level - SIRT1 expression data in human skin we report here – not the least since effects of melatonin mRNA steady-state levels may not be translated into protein level changes following treatment).

The changes are relatively small 10-20% in Fig 1 & 2.  Can these really result in significant changes in skin appearance?

The statistical analysis uses Mann-Whitney, which assumes each measure is independent.  However, the data is 15 samples drawn from only 3-4 cultures, so each data point is not independent. There are other methods to analyze such data and they should be used.

In our study, we carried out regression and correlation analysis to investigate how data points are dependent. Moreover, we also checked normality and lognormality tests (namely D’Agostino and Pearson, Anderson-Darling, Shapiro-Wilk, Kolmogorov-Smirnov), which indicated non-parametrical distribution of variables. Based on the results, we decided to use the corresponding test, Mann-Whitney or Student’s t- test, since we are comparing the vehicle treated samples with 100 or 200uM treated samples. Moreover, our previous data were derived from 15 samples from 3-4 independent donors. Since each individual skin fragment represents its own biological microcosmos, each of which has distinct growth conditions and characteristics, it is plausible to argue that each data point is indeed independent. However, the sample size has now been substantially increased to 5-6, derived from two additional donors, and all data have been statistically re-analyzed as delineated above. This confirmed the robustness of our previous conclusions.

The changes observed in our study may look relatively small if compared to cell culture data in vitro. But one should not forget that these reflect 10-20% changes within a whole human organ that is constituted of multiple different cell types, and a relatively short 6-day treatment period. Thus, it is reasonable to expect that the subtle, but significant effects observed already during such a relatively short treatment window would be maximized upon long-term treatment during a clinical melatonin study, e.g. over several months.

These statistical analysis considerations have now been delineated in the revised M+M section – p.14 - and the above arguments have also been integrated into the revised Discussion – p.8.

The authors should note that the doses used in this study are unrealistically high. Melatonin has a short half-life of 1-2 hrs in vivo and is cleared by the liver.  Even high-dose oral use results in a few hours of serum levels that are 10 to 100-times lower than the doses used here.

Thank you for raising this very important point and for thus encouraging us to clarify this issue. It is correct that even high-dose oral melatonin results only in a few hours of serum levels that are still much lower than the doses used here, largely due to rapid liver metabolism. Previous studies have indeed repeatedly demonstrated that, high doses of melatonin are required to obtain phenotypic effects [16,18,28–30] for example, in normal human epidermal melanocytes [17]. Indeed, these could not be realistically reached with oral administration due to liver metabolism. However, the reviewer does not seem to take into account that very high intracutaneous melatonin levels can easily be reached after topical application of this highly amphiphilic molecule – see above, response to Rev. #2.

Our study was designed to provide a proof-of-concept whether or not high-dose melatonin can indeed exert anti-aging effects in intact human skin. A ‘systemic’ mode of administration, i.e. melatonin addition to the medium, was chosen so as to guarantee optimal melatonin access to the tissue. If our findings are translated into a clinical use of melatonin for anti-skin aging purposes, the most logical mode of application would be a topical one so that high-dose melatonin can be administered directly to the target organ without massive loss of activity due to liver metabolism. We have now clarified this explicitly in the revised Discussion – p.8.

Finally, given that human skin itself is an important extra-pituitary site of melatonin synthesis [31–33], the actual melatonin level that, for example, a melatonin-producing epidermal or hair follicle keratinocyte actually is exposed to in situ remains essentially unknown. It is plausible to assume that this level could may be very high. Thus, it is conceivable that the doses tested here may actually reflect physiological, intracutaneous melatonin levels more closely than the reviewer suspects.

Round 2

Reviewer 2 Report

Comments and Suggestions for Authors

I have no comments.